# The Benefit of Collaboration in the North European Electricity System Transition—System and Sector Perspectives

**Lisa Göransson \*, Mariliis Lehtveer 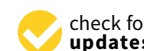, Emil Nyholm, Maria Taljegard and Viktor Walter**

Department of Space, Earth and Environment, Energy Technology, Chalmers University of Technology, 412 96 Gothenburg, Sweden; mariliis.lehtveer@chalmers.se (M.L.); emil.nyholm@chalmers.se (E.N.); maria.taljegard@chalmers.se (M.T.); viktor.walter@chalmers.se (V.W.)
\* Correspondence: lisa.goransson@chalmers.se

**Abstract:** This work investigates the connection between electrification of the industry, transport, and heat sector and the integration of wind and solar power in the electricity system. The impact of combining electrification of the steel industry, passenger vehicles, and residential heat supply with flexibility provision is evaluated from a systems and sector perspective. Deploying a parallel computing approach to the capacity expansion problem, the impact of flexibility provision throughout the north European electricity system transition is investigated. It is found that a strategic collaboration between the electricity system, an electrified steel industry, an electrified transport sector in the form of passenger electric vehicles (EVs) and residential heat supply can reduce total system cost by 8% in the north European electricity system compared to if no collaboration is achieved. The flexibility provision by new electricity consumers enables a faster transition from fossil fuels in the European electricity system and reduces thermal generation. From a sector perspective, strategic consumption of electricity for hydrogen production and EV charging and discharging to the grid reduces the number of hours with very high electricity prices resulting in a reduction in annual electricity prices by up to 20%.

**Keywords:** electrification; sector coupling; integration of VRE; electricity system transition; electricity system modeling; electric vehicles; steel industry; residential heat supply

---

## 1. Introduction

In November 2018, the European Commission presented a strategy for an energy system with net-zero greenhouse gas emissions [1] in order to offer a pathway consistent with the Paris Agreement [2]. In the strategy, the cooperation between sectors is highlighted as a key enabler of the transition of the European energy system and electrification of sectors in the energy system is a part of all scenarios brought forward, implying an increase in annual electricity demand of 35–150% by the year 2050 compared to today. Lechtenböhmer et al. [3] suggest that an extensive electrification of basic industry in EU-28 could increase the electricity demand by 1590 TWh/year, out of which 1200 TWh/year would be in the form of hydrogen. Connolly [4] estimates that if heat pumps are deployed to supply household heating and cooling demands the total electricity consumption in European countries would increase by 30–80% compared to today.

In the strategy presented by the European Commission, the European electricity system is mainly supplied by renewable (wind, solar, hydro, and biomass-based power) and nuclear power. This is in line with recent scenarios, developed by the International Energy Agency, for a clean global energy transition to meet the global climate targets [5]. In large parts of northern Europe, wind power is

the technology offering electricity associated with low or no carbon dioxide emissions to the lowest levelized cost [5]. If the cost of solar power continues to decrease, as expected [5], it could provide electricity with a relatively low levelized cost of electricity in the central and southern part of Europe. However, due to the variability of wind and solar power generation, their value to the electricity system is reduced as their share in the electricity system increases [6]. Variation management strategies (VMS) can mitigate the decline in value of wind and solar power to ensure cost-competitiveness despite high shares of wind and solar in the electricity system [7]. Variation management can be provided by dedicated storages (e.g., stationary batteries, hydro power reservoirs, or hydrogen storage) but also be found in the linkage to other sectors such as the heat, transport, and industry sectors [8,9]. Previous work investigating the impact of electrification on the electricity system has primarily investigated the impact of the heat sector (e.g., [10,11]) or the transportation sector on the electricity system (e.g., [12–14]). Some studies include a parallel electrification of several sectors. Baruah et al. [15] use a combination of soft linked demand and supply side models to investigate the electrification of heat and transportation in Great Britain and conclude that the deployment of demand-side flexibility can reduce the need for thermal complements to offshore wind power significantly. The impact of sectorial linkage on investments in the electricity system has also been analyzed in optimization models. Lund et al. [16] investigate a transition pathway for the Nordic–Baltic energy system and find that there is enough low-cost flexibility in the connection between the Nordic power market and the local heat, gas, and transport markets to allow for large-scale renewables integration. Brown et al. [17] investigate the role of sector coupling on a decarbonized European electricity system and find that battery electric vehicles, power-to-gas technologies and long-term storages make a significant contribution to reducing wind and solar variability and total system cost.

The consequences of an electrification for new electricity consumers is less investigated. McPherson and Tahseen [18] show how the profitability of energy storage increase with the share of wind and solar power in the system. Lund et al. [16] recommend that policy makers assure that market signals for flexibility reach end users and a dynamic taxation of electricity in order to activate cost-efficient flexibility from sector coupling. In many sectors, electricity is one of several potential energy carriers in a carbon constrained world. In addition, there are often options in how to electrify, i.e., how to dimension infrastructure and storages determining at which rates electricity can be consumed. In order to be able to make an informed decision on whether to consume electricity and, if so, where and when to consume it, potential electricity consumers need support from the energy system research community. The size of investments required in the transition from fossil fuels implies that the development of the energy system over decades become relevant and choices which are favorable across many different scenarios should be preferred.

The aim of this paper is to investigate the impact of combining electrification of the industry, heat and transport sector with flexibility provision on: i) electricity system investments and operation and ii) the electricity price perceived by the electricity consuming sectors. To consider the impact of variation management provision on investments and operation of the power system is computationally challenging. Previous work has chosen to tackle the computational challenge by limiting the temporal resolution [16] or confining the study to a national geographical scope [19,20] or by limiting the set of generation technologies [21]. This work contributes by deploying a novel semi-heuristic approach to the optimization problem allowing for a large geographical scope together with a 3-hour time resolution and a large set of generation technology options so that sector coupling can be investigated throughout the transition from the present North European electricity system to a decarbonized alternative of the same and by quantifying the differences between electrification options for new electricity consumers in the industry, heat, and transportation sector in this transition.

## 2. Materials and Methods

### 2.1. Model Description and Scope

The value of sectorial collaboration for the North European electricity system is investigated using a semi-heuristic, cost-minimizing electricity system investment model called Hours-to-Decades (H2D). In the H2D model, investments in generation, transmission and storage capacity are made based on solving the capacity expansion problem (i.e., minimize investment and operational costs to meet the demand for electricity) iteratively for 2-week segments with a 3-hour time resolution. The temporal scope of the segments is chosen to capture wind power variations which frequently occur in the range of 8 days at the hub height of modern turbines [22] and load variations with a regular weekly pattern. The cost-optimal capacity investments of the 2-week segments are used to determine the distribution of investment costs across the 2-week segments the following iteration, such that 2-week segments with large investments in a certain type of capacity get to pay a larger share of the investment cost for that capacity than 2-week segments with less investment in the same. Through iterative solves, the 2-week segments converge in terms of generation, storage and transmission capacity expansion. A mathematical formulation of the model is given in Appendix C and a comprehensive model evaluation can be found in [23].

The H2D model include several aspects of variation management. Thermal cycling is considered by separating between heated capacity available for electricity generation and actual electricity generation as suggested by [24] and evaluated in [25]. In the H2D model, the cost of starting thermal generation depends on the state of thermal generation the 2-week segment before (as given by previous iteration). There is a value to heated thermal capacity at the last time-step of operation which depends on the state of thermal generation the following 2-week period (also from previous iteration). Investments in hydrogen storage, two types of heat storage and two types of stationary battery storages are possible, and the operation of the storages is dispatched such that the storage level the last time-step in the 2-week period equals the storage level the first time-step of the same 2-week period. In addition, variation management through electrification of the steel industry and passenger vehicles, as well as, replacement of fossil fuels for residential heating is included as described in Sections 2.2–2.4. The full range of generation and storage technologies considered in this study, and their properties, can be found in Appendix B. A broad assessment of the role of various generation technologies in the north European electricity system transition is prioritized over a representation of current political climate in the modeled countries. Thus, investments in nuclear power and fossil power with carbon capture and storage is allowed in all regions investigated.

The geographical scope considered in this work includes Denmark, Estonia, Germany, Ireland, Latvia, Lithuania, the Netherlands, Norway, Poland, Sweden, and the UK subdivided into 12 regions to represent major transmission bottlenecks. Figure A1 gives the geographical scope together with the nomenclature of the regions applied in this work. Trade across the geographical scope is allowed and investments in new transmission capacity among regions is possible. The temporal scope of the study is three decades (2025–2055). Over the time period investigated there is an exogenous increase in cost of carbon dioxide, a reduction in investment costs for solar photovoltaics (PV), wind power, and flexibility measures and gradual efficiency improvements for thermal generation. Since years which are close to each other in time will present very similar solutions, parameter values corresponding to 2030, 2040, and 2050 are chosen to represent respective decades. The cost of carbon dioxide is assumed at 40 EUR/ton in 2030, increasing to 100 EUR/ton in 2040 and finally reaching 400 EUR/ton in 2050 to incentives a complete removal of carbon dioxide emissions from the electricity system. The existing powerplant fleet, as given by the continuously updated Chalmers power plant database [26], is taken as a starting point, and generation capacity is gradually phased out as present power plants reach their technical end of life. Details on costs assumed for different generation technologies are included in Table A1. The electricity demand for current applications is assumed to remain at today's level for Northern Europe. Electricity demand from electrification of the steel industry and passenger

vehicles together with replacement of fossil fuels with electricity for heating (using heat pumps with a coefficient of performance of 3) is added to this demand as explained in Sections 2.2–2.4. Overall, this electrification results in an additional electricity demand of around 900 TWh per year by 2050, corresponding to an increase in electricity demand of around 50%.

The cost of EVs and local infrastructural costs associated with the new electricity and heat demands, such as electricity or district heating distribution grids, are outside the scope of this paper.

## 2.2. Transportation Sector

Passenger vehicles are assumed to be subject to a gradual electrification resulting in a full electrification of all passenger vehicles by 2050. Table 1 gives the annual electricity demand for passenger vehicles for the regions and years considered in this work, based on an electricity demand of 0.17 kWh/km and an annual driving distance of 15,000 km per year [27]. The passenger vehicles are assumed to have a battery corresponding to 30 kWh of active battery capacity. Charging infrastructure is assumed to be available at all locations where vehicles are parked for six hour or more (i.e., at home and at work) and vehicle charging, and discharging to the grid, is limited to 3.7 kWh per hour and vehicle.

**Table 1.** Annual electricity demand for transport based on an electrification rate of the present passenger vehicle fleet of 50% by 2030, 70% by 2040, and 100% by 2050. An electricity demand of 0.17 kWh/km is assumed and an annual driving distance of 15,000 km per year [27]. See Appendix A for region names.

| Region | 2030 [GWh/year] | 2040 [GWh/year] | 2050 [GWh/year] |
|--------|-----------------|-----------------|-----------------|
| SE_N   | 650             | 920             | 1300            |
| SE_S   | 8400            | 12,000          | 17,000          |
| DE_N   | 33,000          | 47,000          | 67,000          |
| DE_S   | 55,000          | 78,000          | 110,000         |
| BAL    | 4100            | 5900            | 8400            |
| PO_S   | 24,000          | 34,000          | 48,000          |
| PO_N   | 9400            | 13,000          | 19,000          |
| IE     | 4400            | 6200            | 8900            |
| NO     | 4000            | 5200            | 6700            |
| FI     | 5200            | 7300            | 10,000          |
| UK_S   | 43,000          | 61,000          | 87,000          |
| UK_N   | 4000            | 5600            | 8000            |

Data for vehicle driving patterns is based on GPS-measured movement of 426 passenger cars in south-west Sweden [27]. The vehicles are randomly chosen and proven to be representative for the vehicle fleet in Sweden [27]. In the model, the movement of the 426 cars is aggregated to form one time resolved demand for all passenger transportation. Thus, the model entails one aggregated vehicle category, where it is assumed that a share of the fleet is parked and a share is out being driven at each time-step. The consequences of aggregating vehicles compared to include individual driving patterns in electricity system models have been analyzed by Taljegard [28] and are found to be small as long as the battery capacity is at least 30 kWh and charging infrastructure is available at all parking longer than 6 hours (e.g., at homes and workplaces ).

## 2.3. Industry Sector

In this work the anticipated electrification of the industry sector [3] is exemplified by an electrification of the steel industry. Carbon dioxide emissions from steel making can be drastically reduced by using hydrogen from carbon dioxide neutral electricity to reduce iron ore in place of coke. The hydrogen demand for steel making is assumed to be continuous so as to minimize investments in machinery downstream of the reduction process step. As such, the steel industry represents any continuous hydrogen consumer. Table 2 gives the annual electricity demand for hydrogen production

in the steel industry if all coke is replaced with hydrogen assuming the continued production on the current level. The hydrogen required per kton steel produced is estimated to be 4 TWh. The transition to hydrogen in the steel industry is assumed to be taken in 2040 by the countries currently running research and development projects in this direction, followed by the other countries in 2050. In total, the electrification of industry in this work requires almost an additional 330 TWh (250 TWh of hydrogen and losses in the electrolyzer) of electricity demand for the regions investigated by 2050.

**Table 2.** Annual hydrogen demand for replacing coke with hydrogen as reduction agent in the steel production. See Appendix A for region names.

| Region | 2030 [GWh$_{H2}$/year] | 2040 [GWh$_{H2}$/year] | 2050 [GWh$_{H2}$/year] |
|---|---|---|---|
| SE_N | 0 | 12,000 | 12,000 |
| SE_S | 0 | 7600 | 7600 |
| DE_N | 0 | 261,000 | 54,100 |
| DE_S | 0 | 124,400 | 124,400 |
| PO_S | 0 | 0 | 20,000 |
| FI | 0 | 0 | 12,000 |
| UK_S | 0 | 0 | 18,800 |

### 2.4. Heat Sector

In Germany and in the UK today, fossil fuels, primarily natural gas, are deployed extensively to serve domestic demand for heating and hot water. In this work, this fossil fuel demand is assumed to be gradually phased out. Furthermore, substantial plans for building refurbishments based on EU reports for cost-optimal levels of energy-performance of buildings [29] are assumed to be realized, resulting in lower domestic space heating and hot water demand. The hourly demand profiles for the space heating demand from the residential buildings sectors in the regions in the UK and Germany are derived using a building energy balance model and archetypal buildings as representatives of the building stock. The building energy balance model is a two-node model which accounts for heat transfer through the building envelope and ventilation, solar irradiation, and internal heat gains. For further description of the building energy balance model see Nyholm [30]. The domestic hot water demand is assumed to be constant over the year. In order to extend the heat load from single buildings to country level, the archetypal building stock descriptions for the current building stock in each modeled country is taken from Mata et al. [31]. Table 3 gives the resulting annual energy demand for replacing fossil fuels for heating in Germany and the UK. In all, the changes in Germany and the UK building stocks results in an additional 365 TWh of heat demand by 2050 (corresponding to 122 TWh of electricity demand if served by heat pumps). A further description of the methodology for deriving the heat load profiles applied in this work is provided in Appendix D. Heat demand in the other countries which is today served by units generating or consuming electricity is also include in the model (based on data from [32]) together with the possibility to invest in combined heat and power plants, heat pumps and electric boilers.

**Table 3.** Annual energy demand for replacing fossil fuels for heating with a replacement rate of 9% by 2030, 50% by 2040, and 91% by 2050. See Appendix A for region names.

| Region | 2030 [GWh$_{heat}$/year] | 2040 [GWh$_{heat}$/year] | 2050 [GWh$_{heat}$/year] |
|---|---|---|---|
| DE_N | 6900 | 38,000 | 69,000 |
| DE_S | 17,000 | 93,000 | 170,000 |
| IE | 370 | 2000 | 3700 |
| UK_S | 11,000 | 61,000 | 110,000 |
| UK_N | 1100 | 6300 | 11,000 |

### 2.5. Scenarios

Two main scenarios are considered in this work; the Collaboration scenario and the No Collaboration scenario. The scenarios differ in how demands for electricity and heat from the transport, industry and heat sector are being integrated in the electricity and district heating systems. Thus, the same energy services are fulfilled in both scenarios. Table 4 summarizes the differences between the scenarios. In the No Collaboration scenario, the new demands for heat and electricity has a predefined temporal distribution (i.e., no load shifting is assumed). EVs recharge their batteries directly whenever they are parked for 6 hours or more. The charging is only limited by the charging capacity, i.e., 3.7 kWh per hour, and does not consider the availability or price of electricity. Electricity required to produce hydrogen for the steelmaking process is in the No Collaboration scenario evenly distributed across the hours of the year, reflecting the preference to operate the shaft furnace continuously. Hydrogen storage is not available for investments in this scenario. Furthermore, in the No Collaboration scenario, fossil fuels for heating is replaced by individual heat pumps located at the individual buildings and operated according to the heat demand profile of the households. The cost of the individual heat pumps is part of the total system cost in the No Collaboration scenario.

**Table 4.** Differences between the two scenarios Collaboration and No Collaboration.

| Sector Coupling Strategy | Collaboration | No Collaboration |
|---|---|---|
| EV charging strategy | Optimized including V2G | Directly after each trip |
| Hydrogen storage | Rock cavern storages available | No hydrogen storage available |
| NG heat replacement | District heating supplied by CHP, EB, or HP | Individual heat pumps |
| Heat storages in DH | Tank, pit storages available | No heat storage available |

In the Collaboration scenario, on the other hand, EV charging is flexible in time for 30% of the vehicles while still meeting the demand for transportation at all times. It is also possible for these EVs to discharge back to the grid (V2G). 70% of the EVs charge as soon as they are parked. Charing infrastructure is assumed to be available to the same extent as in the No Collaboration scenario. The cost of the vehicles and a potential reimbursement to the EV owner for optimizing their charging is outside the system boundary of this work. As for the steel industry, there is an option to make additional investments in electrolyzer capacity and to make investments in hydrogen storage in the Collaboration scenario. These investments are made so as to minimize total system costs. Natural gas for heating is in this scenario replaced by district heating. The heat demand in district heating can be met by combined heat and power (CHP), electric boilers (EB) and heat pumps (HP). There is also a possibility to invest in tank and pit storages.

## 3. Results

### 3.1. System Planner Perspective

We find that collaboration between the electricity system and an electrified steel industry, passenger vehicles and household heat supply can reduce total system costs by 8% under the assumptions made in this work. All sectors contribute significantly to the total system cost reduction (i.e., strategic electricity consumption in EVs, by the steel industry, and for space heating are individually responsible for a total system cost reduction of 3.5%, 1.5%, and 3%, respectively). However, the north European electricity system composition is found to be dominated by wind power irrespective of whether sectorial collaboration is available or not. Figure 1 shows the annual electricity generation in northern Europe for the two scenarios and the years investigated. In 2050, sectorial collaboration increases the share of the demand supplied by wind power from 61% to 63% and by solar PV from 16% to 19%.

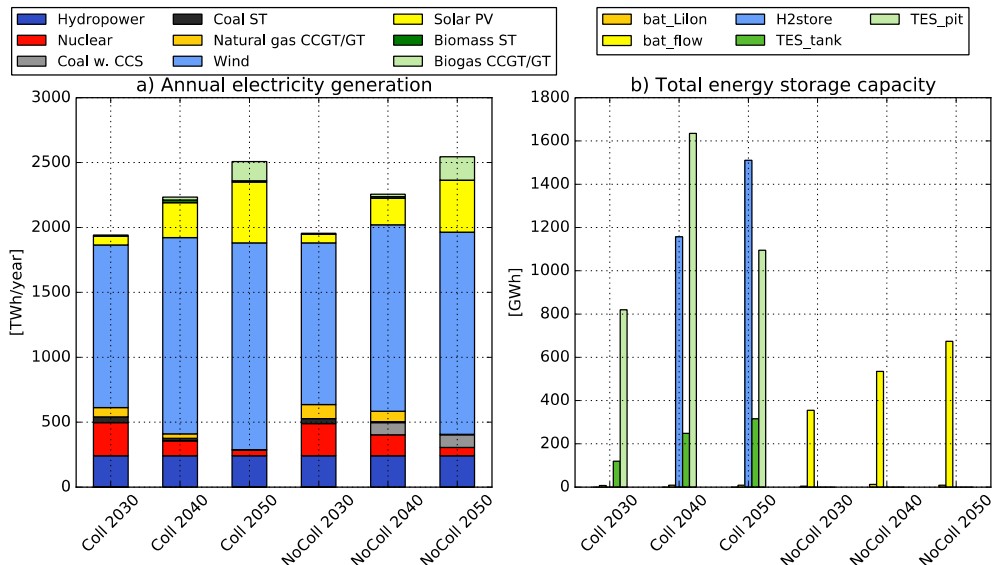

**Figure 1.** The annual electricity generation (**a**) and the total energy storage capacity (**b**) for the two scenarios (Sectorial Collaboration and No Sectoral Collaboration) for the years 2030, 2040, and 2050. Coll—collaboration, CCS—carbon capture and storage, CCGT—combined cycle gas turbine, ST—steam turbine, H2store—hydrogen storage, TES—thermal energy storage, bat_Lion—stationary lithium-ion batteries, bat_flow—stationary flow batteries.

Without sectorial collaboration thermal generation supply a larger share of the electricity demand. The transition to renewable electricity generation is also slower, and the largest impact of sectorial collaboration is achieved in 2040, when a large share of the flexibility from the collaboration is assumed to be in place but the cost of carbon dioxide remains moderate. In 2040, thermal generation is represented by natural gas combined cycle and open cycle gas turbines (46 TWh/year higher without collaboration), bio-blended coal power with carbon capture and storage (CCS) (90 TWh higher without collaboration), and nuclear power (47 TWh higher without collaboration).New investments in nuclear power and bio-blended coal power with CCS is mainly located to the regions with very high electricity consumption, i.e., DE_S and UK_S. Bio-blended coal power with CCS with good load following properties (see Table A1) but slightly higher investment cost are favored over less flexible such.

The largest difference between the scenarios investigated is how the variability is managed. In the No Collaboration scenario, the high share of varying renewables is achieved through large investments in battery capacity together with operation of combined-cycle and open-cycle gas turbines. Figure 1b gives investments in energy storage for the scenarios considered. In the No Collaboration scenario around 700 GWh of stationary batteries are invested in by 2050. This can be compared to the 5 TWh of batteries expected to be required in the transport sector in the same regions by mid-century. In the electricity system, batteries with low energy storage costs and lower c-factor and efficiency (i.e., flow batteries) is preferred to batteries with a higher energy storage cost but higher c-factor and efficiency (i.e., lithium-ion batteries). The flow batteries are here assumed to have a 70% efficiency, and, due to a large number of cycles, the total electricity demand in the No Collaboration scenario is 50 TWh higher than that of the Collaboration scenario by 2050. Flexible generation, in the form of combined-cycle gas turbines, support the electricity system during periods of prolonged low wind power generation. Natural gas fired combined-cycle gas turbines dominate up until 2040, but in 2050 these are replaced by their bio-gas fired counterpart. By 2050, biogas-based electricity generation is 32 TWh higher per year in the No Collaboration scenario compared to the Collaboration scenario.

With sectorial collaboration, investments in stationary battery capacity in the electricity sector is negligible (8 GWh in total). Instead, variability is managed through storing electricity in EV batteries and in dedicated hydrogen and heat storages. Figure 2 shows the operation of the storages together

with electricity generation and electricity demand for the region UK_S two weeks in early spring (weeks 6 and 7). UK_S is one of the regions with largest electricity demand, but it has at the same time very limited possibilities for transmission and is thereby forced to manage the majority of the variability locally. Figure 2a shows the electricity generation. Wind power produce more in the first week than in the second week and solar power is subject to large variations in diurnal peak due to difference in cloud coverage between the days. Thermal power has a relatively stable production pattern. Biogas combined cycle generation is lower during a pronounced peak in wind power generation which last around 2 days. Nuclear power is unaffected by the wind and solar variations in this 2-week period. The variations are instead managed by the temporal distribution of the electricity consumption, given in Figure 2b. The most prominent variation management is performed by EV charging (i.e., EV demand in Figure 2b), which coincides with solar peak generation. With 30 % flexible vehicles, 11 M cars in UK_S are optimized to charge and discharge strategically. This corresponds to a charging capacity of 41 GW (in the unlikely event all vehicles are parked) and a storage capacity of 330 GWh with the assumptions on battery size and connection capacity made in this work. Figure 2b illustrates how the charging and storage capacity of the EVs to a large extent is utilized to accommodate solar PV. EVs discharging to the grid (V2G, negative values in the EV demand in Figure 2b) occurs during the most critical hours when wind and solar production is low but demand for electricity is high (cloudy, still days). Also, the electrolyzers supply the system with variation management by interrupting electricity consumption for hydrogen production during these critical hours. By reducing the need for peak generation complementing wind and solar power the cost-competitiveness of VRE relative other generation technologies is enhanced. Figure 2c shows how the hydrogen storage is filled slowly during the first week with medium to good wind conditions as the electrolyzer operates at a rated power which is slightly higher than what is required to meet the nominal demand for hydrogen. The storage is then discharged over a couple of critical days during which the electrolyzer is taken out of operation. Figure 2c also illustrates the operation of heat storages, where the tank is charged and discharged to meet the diurnal variations in heating demand while maintaining the heat pump capacity low. The heat pump (Figure 2b) is operated almost continuously, with only slight reduction in electricity consumption during critical hours. The pit storage is slowly charged during hours of medium to good wind conditions to discharge during low wind events.

In both the Collaboration and No Collaboration scenario the transmission grid primarily serves the purpose of exporting electricity from regions with good conditions for wind power generation but low local electricity demand to regions with extensive electricity demand. DE_S is the main importing region, receiving 300 TWh/year in 2050, followed by UK_S and SE_S which each import around 60 TWh/year in 2050 for both scenarios investigated. BAL and NO are the main exporting regions (120 TWh/year by 2050), followed by UK_N, SE_N, and PO_N (60 TWh/year in 2050). Through a strategic temporal distribution of import and export, variations can be managed while resources are redistributed geographically. For example, EV charging in UK_S supports solar PV integration in DE_S (see for example import peaks at hour 61 and 255 in Figure 2a) while hydropower rich NO and UK_N support UK_S during low wind events. Export from regions with extensive hydropower capacity (i.e., SE_N, UK_N, and NO) to neighboring regions during hours of high net load is a recurring feature which mitigate the need for peak generation in both the Collaboration and No Collaboration scenario.

Just as trade with regions with vast amounts of hydropower, collaboration with the other sectors support the electricity system during low wind events and the need for peak capacity (i.e., biogas open-cycle gas turbines) is reduced from 13 to 4 GW in northern Europe in the Collaboration scenario relative the No Collaboration scenario by 2050. At the same time, competitiveness of wind and solar power over other generation technologies is increased, resulting in reduced investments also in nuclear and CCS relative the No Collaboration scenario. However, only EVs have the ability to give value to electricity during low or negative net load resulting in a relatively high amount of annual curtailment, 94 TWh (4% of total generated electricity) per year in 2050 for northern Europe as a whole. This is 38 TWh more curtailment than in the No Collaboration scenario.

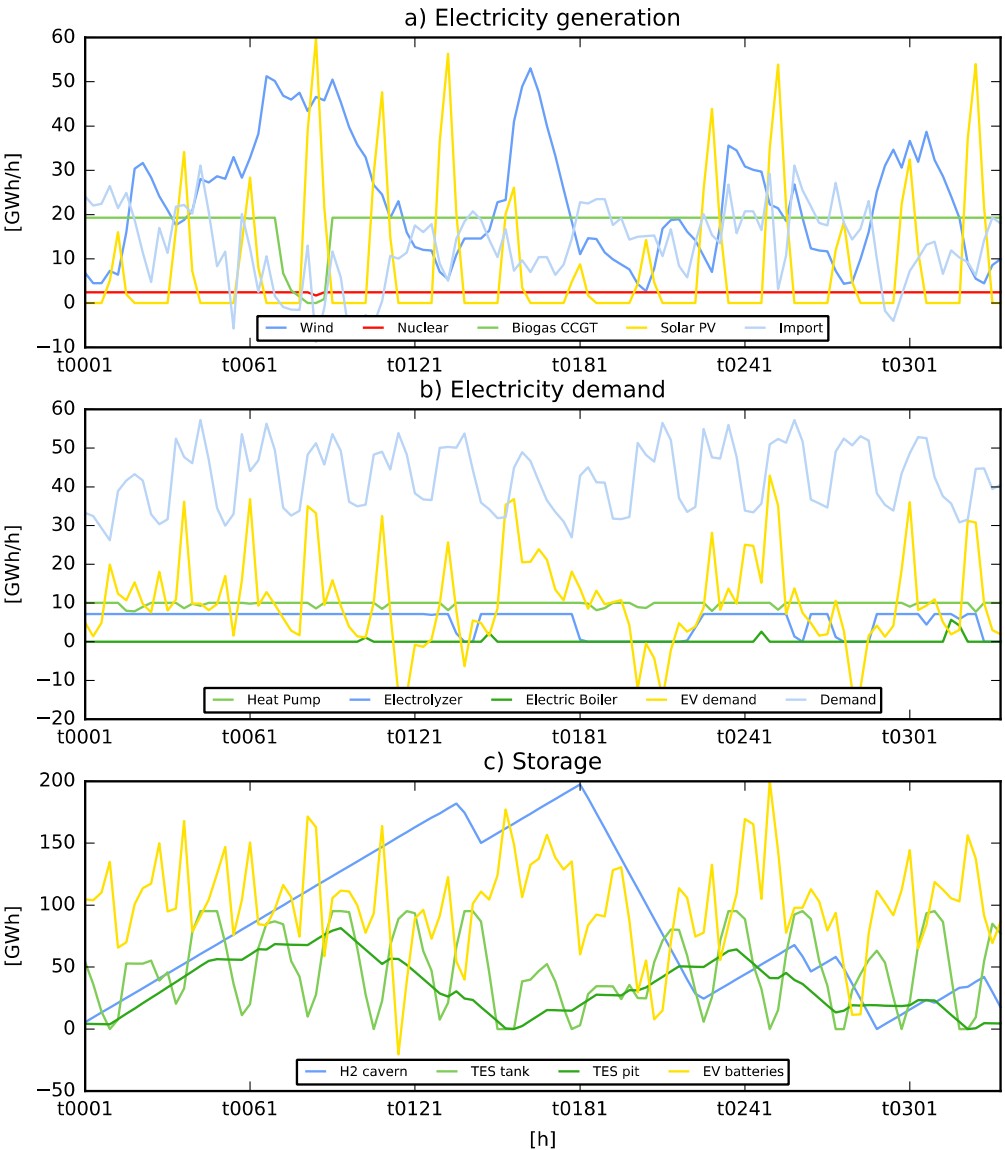

**Figure 2.** (**a**) Electricity generation; (**b**) electricity demand; and (**c**) energy storage for region UK_S (England and Wales) in weeks 6 and 7 of year 2050.

*3.2. Sector Perspective*

With the increased share of varying renewable generation over the years investigated, the number of hours with (very) low and high electricity prices increase whereas the hours with intermediate electricity prices are reduced. Because of the different nature of the strategies to manage variations in the Collaboration and the No Collaboration scenario, price formation between the scenarios is also different. Figure 3 gives the price duration curves for UK_S (England and Wales) for the years investigated. The regular deployment of biogas combined-cycle units to manage prolonged high net load events in both scenarios result in many hours with high electricity prices. In the Collaboration scenario, demand for electricity for hydrogen, heat generation and EV charging can be moved from very high net load hours, reducing the need for biogas open-cycle generation and the number of hours with very high electricity prices. Meanwhile, stationary batteries in the No collaboration scenario can efficiently absorb electricity during low net load events, thus reducing the number of hours with very low electricity prices. The average annual electricity price is therefore lower with sectorial collaboration. Figure 4a gives the reduction in average annual short-term marginal cost of electricity, the electricity price on an energy-only market, from collaboration for the regions and years investigated.

The average annual electricity price is 9–21% lower (5–14 EUR/MWh depending on region) with sectorial collaboration in the year 2050.

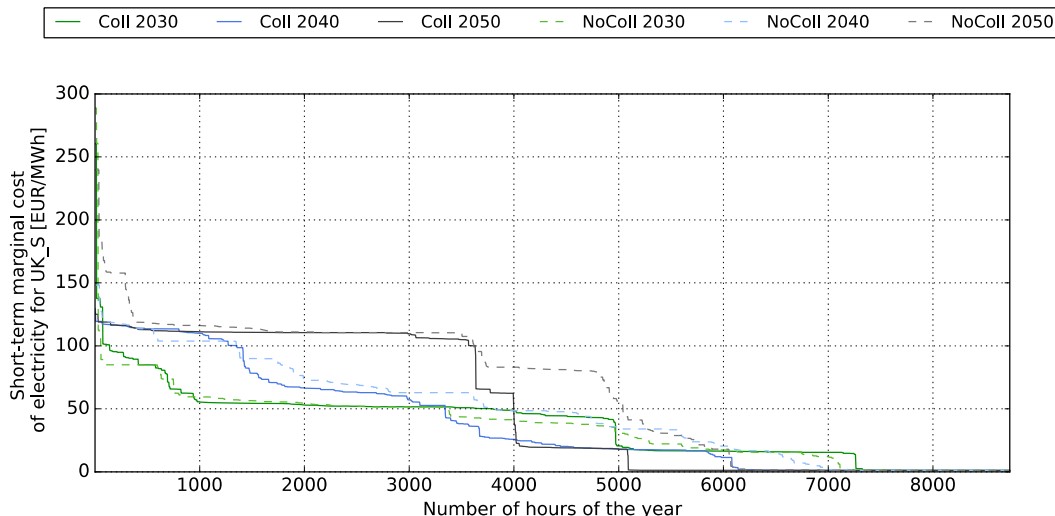

**Figure 3.** Duration of short-term marginal cost of electricity for region UK_S (England and Wales) for the Collaboration (solid lines) and No Collaboration (dashed lines) scenarios and the years investigated. Coll—collaboration.

Whereas the average annual electricity price is relevant to all consumers with an electricity demand evenly distributed over the year, or with an electricity demand uncorrelated to the net load, the electricity price perceived by the heating, transportation, and industry sector needs to be evaluated separately. Uncontrolled electricity demand for heating and transport, applied in the No Collaboration scenario, has a certain correlation with the demand for electricity. In the Collaboration scenario, the demand for electricity in heating, transportation, and industry is manipulated to manage variations in net load.

Figure 4b gives the reduction in consumption weighted electricity price for hydrogen production for industry with sectorial collaboration compared to without collaboration. In the No Collaboration scenario, electricity consumption for hydrogen production is assumed to follow a hydrogen demand which is evenly distributed over the year. Thus, the electricity price for hydrogen production equals the average annual electricity price for this scenario. With sectorial collaboration, a slight additional investment in electrolyzer capacity and investment in hydrogen storage enables hydrogen production to move away from high cost hours. Thus, the consumption weighted electricity price for hydrogen is slightly lower than the average annual electricity price, resulting in a 7–17 EUR/MWh reduction in the electricity price perceived by the hydrogen consumer in the year 2050 in the Collaboration scenario. The hydrogen storages are typically sized to supply the steel industry with hydrogen for two days, with Finland as outliers with 1 day of storage, respectively. The electrolyzers are dimensioned to fill the storage in five days of operation at rated power (while also supplying the industry with hydrogen), with electrolyzers in Finland as exceptions requiring 7 days to fill the storage. Table 5 gives the annualized investment cost of additional electrolyzer capacity and hydrogen storage capacity together with the income from that flexibility as a result of a reduced electricity price and return on investment (i.e., taken as savings over costs) as perceived by the consumer calculated on numbers for year 2050.

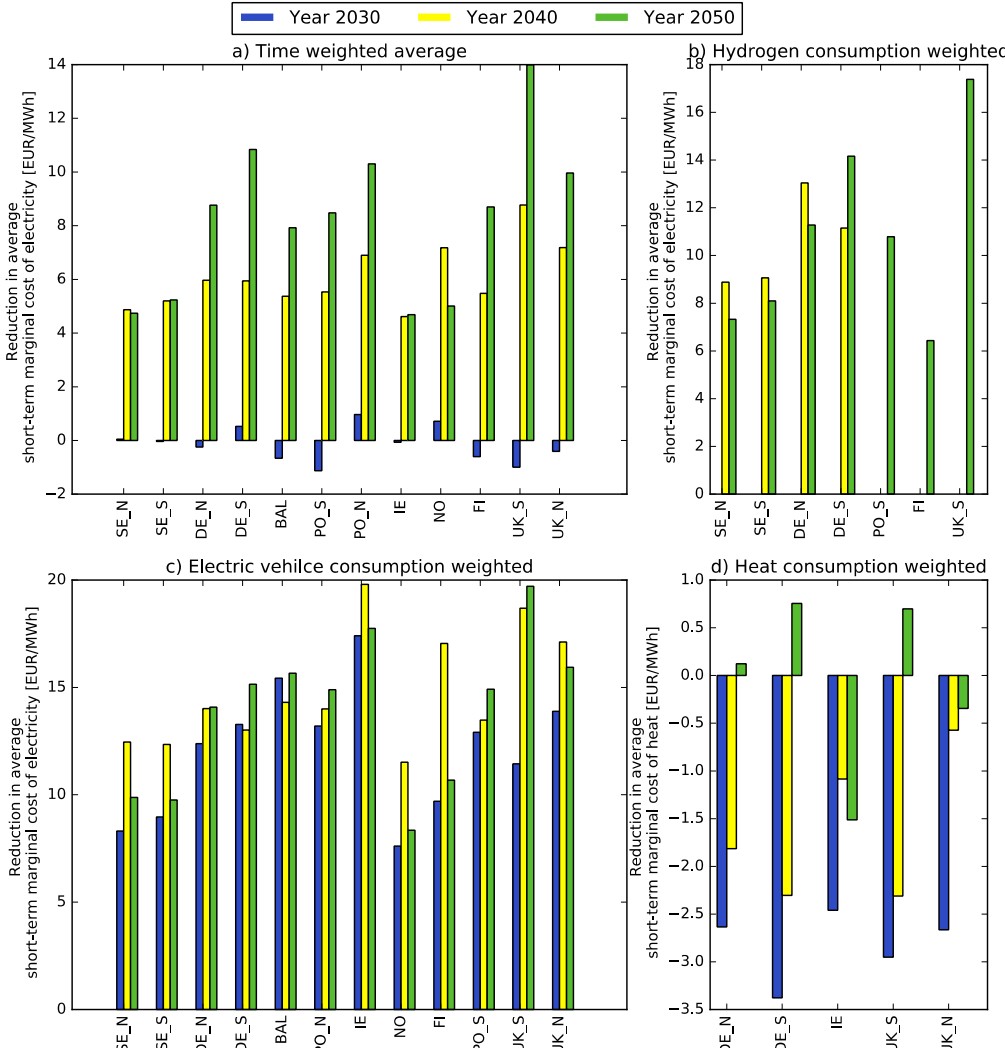

**Figure 4.** Reduction in average annual short-term marginal cost of electricity from collaboration relative to if no collaboration is achieved as perceived by: (**a**) base load; (**b**) hydrogen producers; (**c**) electric vehicle owners who charge and discharge strategically; and (**d**) heat producers.

**Table 5.** Relation between annual costs and savings related to flexible hydrogen consumption where costs correspond to annualized investment cost for hydrogen storage and electrolyzer capacity (in addition to capacity needed to serve the hourly hydrogen demand from industry) and savings correspond to the reduced cost for electricity used in hydrogen production in the Collaboration scenario as compared to the No Collaboration scenario. See Table A2 for cost assumptions and Figure 4b for reduction in electricity price from collaboration.

| Regions | Cost of $H_2$ Flex (MEUR/year) | Savings from Reduced Electricity Price (MEUR/year) | Return of Investment |
|---|---|---|---|
| SE_N | 64 | 49 | 0.8 |
| SE_S | 44 | 37 | 0.9 |
| DE_N | 640 | 813 | 1.3 |
| DE_S | 1793 | 2350 | 1.3 |
| PO_S | 221 | 288 | 1.3 |
| FI | 120 | 103 | 0.9 |
| UK_S | 458 | 825 | 1.8 |

Figure 4c shows the reduction in consumption weighted electricity price for electricity for EV charging from sectorial collaboration relative to if no collaboration is achieved. Just like electricity for hydrogen, the electricity price perceived by EVs is close to the average annual electricity price in the No Collaboration scenario. With optimized charging with V2G, however, the average annual electricity price perceived by EVs is up to 8 EUR/MWh lower than the time weighted average electricity price for the same scenario. The large impact of unlocking the EV flexibility on the perceived electricity price is a result of the high charging capacity available, resulting in the peaky charging behavior illustrated in Figure 2b. Sectorial collaboration, including optimized charging with V2G, reduces the average annual electricity price for vehicle charging with up to 10–20 EUR/MWh in 2050. It is found that strategic integration of passenger EVs alone reduce the total system cost by 3.5%. Running the Collaboration scenario with all vehicles charging and discharging strategically would further reduce total system costs by 3%.

The demand for heat is mainly served by heat pumps in both scenarios investigated, either small-scale heat pumps at the home location or large-scale heat pumps feeding the district heating system. In the collaboration scenario, the connection to the district heating system together with the possibility to invest in thermal storage tanks and pits enables a shift in time between supply and demand. This possibility is primarily utilized to reduce peaks in heat load such that the demand for heat can be served with a smaller investment in heat generation capacity in the district heating system. In the year 2050, 230 GW heat pump capacity is installed in the No Collaboration scenario, whereas this is reduced down to 80 GW with collaboration. Heat pumps are thus operated at rated power during periods of high heat demand. Since the electricity price is mainly governed by the availability of wind and solar power, neither the electricity consumption for heating in the No Collaboration or Collaboration scenario correlate with the electricity price. The collaboration between the electricity system and the district heating system thus mainly reduce investment costs rather than operational costs of the system, and while the consumption weighted heat price experienced by the heat consumer is slightly reduced in some regions it is slightly increased in others. Accounting for the possibility of storing heat between seasons could reduce the consumption weighed heat price but would have low impact on total system configuration and costs.

## 4. Discussion

### 4.1. Model Limitations

This work has an electricity system and sector focus. This is only a part of a larger transition involving investments in vehicles and infrastructure outside the scope of this work. The comparison between replacing natural gas with district heating or individual heat pumps provided in this work needs to be extended with a comparison of costs between district heating systems and local distribution grid expansion for a fair evaluation between options.

The method applied here is suitable for electricity systems dominated by wind variations, as is expected in the electricity system in northern Europe. However, for the heating systems there are important seasonal variations for which thermal energy storage, in particular pit storages, could be valuable. These values are not captured in this work. The limited foresight nature of the method impact electricity prices in the presence of large-scale storages, which now only flatten electricity prices across 2-week periods rather than across the whole year (which happens in the case of large-scale storages in perfect foresight models). It can be argued that limited foresight storage planning is closer to reality due to limitations in forecasting. However, the timespan of 2 weeks is chosen to represent wind variation management well rather than accurately assess forecast limitation.

### 4.2. Result Discussion

The integration of wind and solar power in the electricity system is not dependent on sectorial collaboration. However, from a system planner perspective sectorial collaboration offers a more robust

transition pathway for the electricity system by reducing the reliance on thermal generation and stationary batteries. Without sectorial collaboration, investments in new nuclear power and coal power with CCS are made in UK_S and DE_S and it is likely that these investments would meet resistance. The Germans have decided against both of these thermal generation options. In addition, the No Collaboration scenario deploy biomass for load following to a greater extent. Biomass is a limited resource needed in the transition of many sectors of the economy, thus its future cost and availability is highly uncertain. Similarly, large volumes of batteries are needed in the transition of the transportation sector and even though this may push for further cost reductions it also implies a risk for resource scarcity and battery factory capacity limitations. From an electricity consumer perspective, it may be less problematic to argue for a sustainable end-product (steel or transportation) with reduced reliance on nuclear power, coal power with CCS and natural gas (model year 2040) or biogas (model year 2050).

Only part of the heat, industry, and transportation sectors are represented in this work. Hydrogen from electrolysis could be utilized in refineries and biorefineries as well as in the production of plastics. Hydrogen is also discussed as on option for heavy transportation. Such development could further increase the flexibility provided by sectorial collaboration. Electrification of industry with batch type processes could add to this flexibility. A transition from natural gas for heating purposes is only included in Germany and the UK. A similar transition is needed in Poland and would primarily add to the base load in these countries in the Collaboration scenario but not provide much additional flexibility.

Increased electrification could also lead to increased competition for the available wind and solar resources as the potential land usage for wind and solar power plants is a social acceptance question. This highlights the importance to include efficiency measures to keep the demand low as well as include the public in the expansion of the electricity system.

This work shows that by consuming electricity strategically when charging electric vehicles and producing hydrogen to the steel industry, the cost of electricity for these consumers can be substantially reduced. Consumers which fail to react to electricity prices due to lack of knowledge, information or resources may find it hard to remain competitive, if the cost of energy is a significant share of their product cost.

## 5. Conclusions

A strategic collaboration between the electricity system, an electrified steel industry, an electrified transport sector in the form of passenger EVs, and residential heat supply enables a faster transition from fossil fuels and reduces total system cost by 8% in the north European electricity system. Collaboration with the different sectors supports the energy system transition in different ways. Strategic integration of EVs increases the value of solar PV and reduces the need for peak generation, whereas strategic production of hydrogen to the steel industry increases the cost-competitiveness of wind and solar power relative thermal generation. Thus, both collaboration with the transport and industry sector reduce the electricity generation in thermal units during the transition to a carbon neutral electricity system. If natural gas for residential heating is replaced by district heating instead of individual heat pumps the heat pump capacity can be reduced by 65 % through the use of thermal energy storages. Benefits should be compared to the cost of building district heating relative a local electricity grid expansion. Strategic consumption of electricity for hydrogen production and EV charging and discharging to the grid reduces the need for peak generation compared to an inflexible consumption of electricity, thereby reducing the number of hours with very high electricity prices as well as the annual average electricity price. EV owners and industries providing flexibility experience greater reductions in annual average electricity price than inflexible consumers. The energy-only market can be expected to give strong economic incentives for EV owners to charge strategically and feedback electricity to the grid. In regions without hydropower, the energy-only market provides incentives to hydrogen consuming industries to invest in flexibility. In regions with hydropower, investments in flexibility may need to be stimulated in order to reach levels desired by a system planner.

**Author Contributions:** L.G., M.L., and V.W. developed the conceptualization of this work. L.G. was responsible for the study design and analysis as well as for the original draft preparation. E.N. and M.T. were responsible for data preparation and methodology for the inclusion of residential heat demand and passenger electric vehicles, respectively. M.L., V.W., E.N., and M.T. contributed with reviewing and editing of the original draft.

**Funding:** This research was funded by MISTRA Carbon Exit, grant number 156414, FORMAS, grant number 2017-01394 and Chalmers Energy Area of Advance.

**Acknowledgments:** Thanks to Ann-Brith Strömberg and MSc Caroline Granfeldt for your support in the mathematical denotation of the H2D model (Appendix A).

**Conflicts of Interest:** The authors declare no conflict of interest. The funders had no role in the design of the study; in the collection, analyses, or interpretation of data; in the writing of the manuscript, or in the decision to publish the results.

## Appendix A

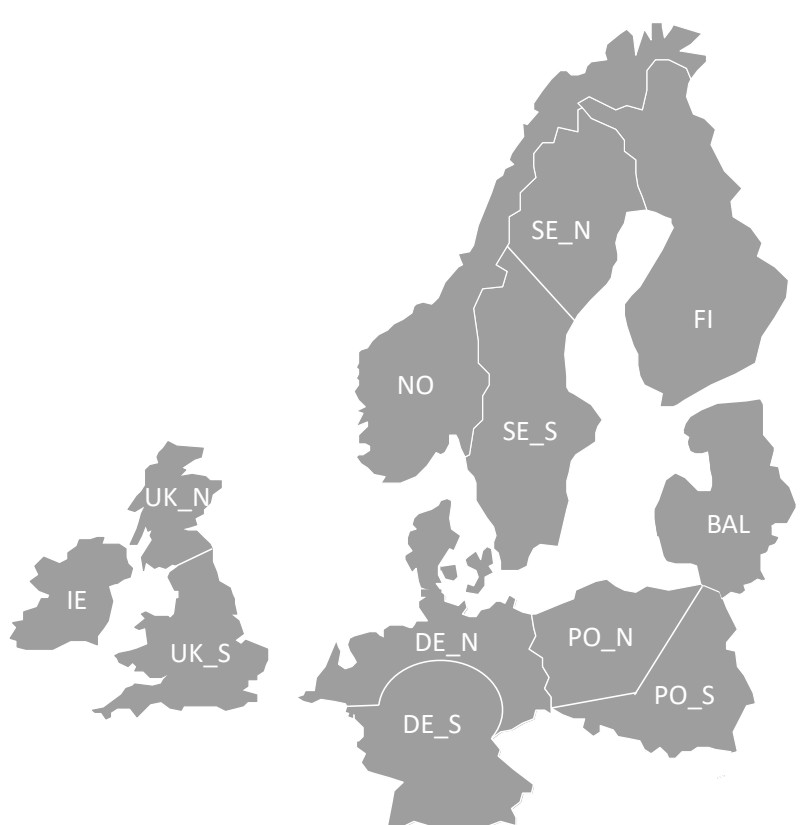

**Figure A1.** Map of regions considered in this work. Northern Europe is subdivided into 12 regions based on major bottlenecks in the transmission grid and key differences in renewable resources.

## Appendix B

Table A1 gives the investment and variable costs for the electricity generation technologies considered in the model. The investment costs and fixed operation and maintenance costs are based on International Energy Agency World Energy Outlook 2016 [33], with the exception of the costs for onshore wind power, which are based on the costs presented by [34] with a yearly learning rate of 0.4%. In the model, annualized investment costs are applied assuming a 5% interest rate. Technology learning for thermal generation is included as gradual improvement in the efficiencies of these technologies, reflected as a reduced variable cost in Table A1. The variable costs listed in Table A2 exclude the cost of carbon dioxide, which varies between years. The cost of cycling thermal generation is not part of the variable cost. Instead, the start-up costs and part-load costs are included explicitly in the optimization. The start-up costs, part-load costs, and minimum load level applied here are based on

the report of [35], in which all the technologies that employ solid fuels use the cycling costs given for large sub-critical coal power plants. The start-up fuel is, however, changed to biogas rather than oil in all bio-based generation in the present work. The cost of carbon dioxide emissions related to starting thermal generation vary from year to year and is therefore not included in the start-up costs in Table A1. The cycling properties of nuclear power are based on the paper by [36], who describe a start-up time of 20 h and a minimum load level of 70%.

Biogas is assumed to be produced through the gasification of solid biomass, with 70% conversion efficiency. The cost of the gasifier equipment is included in the form of 20 €/MWh added to the fuel cost, rather than being incorporated into the investment cost of the biogas technologies, since biogas is storable, which means that the gasifier equipment may attain a much higher number of full-load hours compared to the power plant consuming the biogas. The total cost of the gasification equipment is taken from [37], and 8000 full-load hours are assumed.

**Table A1.** Costs and technical data for the electricity generation technologies. The variable costs and start-up cost are for year 2030 and exclude costs of $CO_2$ emissions.

| Technology | Investment Cost 2030 [M€/MW] | Investment Cost 2050 [M€/MW] | Variable Costs [€/MWh] | Fixed O&M Costs [k€/MW,yr.] | Life-Time [yr.] | Minimum Load Level [Share of Rated Power] | Start-Time [h] | Start Cost [€/MW] |
|---|---|---|---|---|---|---|---|---|
| Coal ST | 1.56 | 1.56 | 17 | 27 | 40 | 0.35 | 12 | 192 |
| Coal CHP | 1.56 | 1.56 | 21 | 27 | 40 | 0.35 | 12 | 192 |
| Coal CCS | 3.00 | 3.00 | 21 | 91 | 40 | 0.35 | 12 | 192 |
| NG CCGT | 0.78 | 0.78 | 39 | 13 | 30 | 0.2 | 6 | 44 |
| NG GT | 0.39 | 0.39 | 64 | 8 | 30 | 0.5 | 0 | 32 |
| NG CHP | 1.01 | 1.01 | 48 | 17 | 30 | 0.32 | 12 | 102 |
| NG CCS | 1.80 | 1.80 | 53 | 35 | 30 | 0.35 | 12 | 192 |
| Biomass ST | 1.86 | 1.86 | 90 | 50 | 40 | 0.35 | 12 | 192 |
| Biomass CHP | 3.15 | 3.15 | 119 | 58 | 40 | 0.35 | 12 | 192 |
| Waste CHP | 6.63 | 6.63 | 7 | 443 | 40 | 0.35 | 12 | 192 |
| Biogas CCGT | 0.76 | 0.76 | 117 | 13 | 30 | 0.2 | 6 | 47 |
| Biogas GT | 0.38 | 0.38 | 195 | 8 | 30 | 0.5 | 0 | 55 |
| Bio-coal CCS (flex) | 3.46 (3.64) | 3.46 (3.64) | 40 | 107 (113) | 30 | 0.35 (0.15) | 12 (6) | 192 (110) |
| Hydropower | 2.06 | 2.06 | 1.0 | 47 | 500 | 0 | 0 | 0 |
| Nuclear | 5.15 | 5.15 | 16.5 | 154 | 60 | 0.7 | 24 | 670 |
| Solar PV | 0.99 | 0.60 | 1.1 | 10 | 25 | 0 | 0 | 0 |
| Onshore wind | 1.33 | 1.23 | 1.1 | 30 | 25 | 0 | 0 | 0 |
| Offshore wind | 3.29 | 2.21 | 1.1 | 100 | 25 | 0 | 0 | 0 |
| Transmission (OHAC) | 0.6 (per km) | 0.6 (per km) | 0.01 | - | 40 | 0 | 0 | 0 |
| Transmission (HVDC) | 0.756 + 0.63 (per km) | 0.756 + 0.63 (per km) | 0.01 | - | 40 | 0 | 0 | 0 |

The wind power generation profiles are calculated for wind turbines with low specific power (200 W/m$^2$), with the power curve and losses proposed by [38]. The wind speed input data are a combination of the MERRA and ECMWF ERA-Interim data for year 2012, whereby the profiles from the former are re-scaled with the average wind speeds from the latter [39,40]. The high resolution of the wind profiles from the ERA-Interim data was processed into wind power generation profiles and put together into 12 wind classes for each region. The wind farm density is set to 3.2 MW/km$^2$ and is assumed to be limited to 10% of the available land area, accounting for protected areas, lakes, water streams, roads, and cities [41].

Solar PV is modelled as mono-crystalline silicon cells installed with optimal tilt with one generation profile for each region. Solar radiation data from MERRA is used to calculate the generation with the model presented by [42], including thermal efficiency losses.

The cost and technical data for variation management is based [43] and presented in Table A2. The hydrogen storage is assumed to be of the large-scale, steel lined cavern type.

**Table A2.** Costs and technical data for the variation management technologies. The costs for electric boilers and electrolyzers are given per MW and the costs of the batteries and hydrogen storage are given per MWh. For the thermal storages there are heat losses in addition to efficiency losses when charging 0.01/240 of the heat content of the storage together with heat corresponding to 0.07/360 of the storage capacity is lost every hour.

| Variation Management Technology | Investment Cost [M€/ MW(h)] | Efficiency [%] | Fixed O&M Costs [k€/MW(h), yr.] | Life-Time [yr.] |
|---|---|---|---|---|
| Battery, Li-ion | 0.15 | 95 | 25 | 15 |
| Battery, Flow | 0.10 | 70 | 13 | 30 |
| Electrolyzer | 0.59 | 75 | 20 | 20 |
| H₂ storage | 0.01 | 100 | - | 50 |
| Heat pump | 1.00 | 300 | 8 | 25 |
| Electric boiler | 0.05 | 100 | - | 20 |
| TES tank | 0.03 | 95 | - | 20 |
| TES pit | 0.004 | 80 | - | 20 |

**Appendix C**

The H2D model consists of two parts: a cost-minimizing electricity system model with a 3-hour time resolution and a temporal scope of two weeks, and a consensus loop sharing information of investment decisions between the two weeks segments. The objective function of the electricity system model for each two-week segment s is expressed as:

$$MIN\ C_{tot} = \sum_{i \in I} \left( \sum_{p \in P} C_p^{inv} \sum_{cl \in CL} \left( C_{i,p,cl}^{share} w_{i,p,cl} \right) + \sum_{p \in P} \sum_{t_{s,r} \in T_s} \left( C_{p,t_{s,r}}^{run} g_{i,p,t_{s,r}} + c_{i,p,t_{s,r}}^{cycl} \right) + \sum_{q \in Q} C_q^{h\_inv} \sum_{i^* \in I \backslash i} \sum_{cl \in CL} \left( C_{i,i^*,q,cl}^{h\_share} h_{i,i^*,q,cl} \right) + \sum_{i^* \in I \backslash i} \sum_{t_{s,r} \in T_s} \left( C_{t_{s,r}}^{exp} e_{i^*,i,t_{s,r}}^{pos} \right) \right), \forall s \in S \tag{A1}$$

where

I is the set of all regions;

P is the set of all technology aggregates;

P^gen is the set of all electricity generation technologies;

Q is the set of technologies for transmission;

S is the set of all two-week periods;

$T_s$ is the set of all time steps in the two-week aggregate s, s∈ and with $|T_s| = |T|$, $\forall s \in S$ and $r \epsilon \{1, \dots, |T|\}$, $t_{s,r} = (s-1)|T| + r$, $\forall\ r \in T$;

CL is the set of cost classes, i.e., the steps in the cost-supply curve;

$C_p^{inv}$ is the investment cost of technology p;

$C_q^{h\_inv}$ is the investment cost of transmission technology q;

$C_{i,p,cl}^{share}$ is the share of the investment cost taken by cost class cl in region i for technology p;

$C_{i,i^*,q,cl}^{h\_share}$ share of investment cost for cost class cl for transmission between regions i and i*;

$C_{p,t_{s,r}}^{run}$ is the running costs of technology p at time t;

$C_{t_{s,r}}^{exp}$ is the cost of transmitting electricity;

$w_{i,p,cl}$ is the positive variable representing investment in generation and storage capacity;

$h_{i,i^*,q,cl}$ is the positive variable representing investment in transmission capacity;

$g_{i,p,t_{s,r}}$ is the positive variable representing electricity generation;

$e_{i,i^*,t_{s,r}}$ is the free variable representing export from i to i*;

$e_{i^*,i,t_{s,r}}^{pos}$ is the positive variable consistent with $e_{i^*,i,t_{s,r}}^{pos} \geq e_{i^*,i,t_{s,r}}$

$c_{i,p,t_{s,r}}^{cycl}$ is the positive variable representing thermal cycling costs.

And the years modelled are omitted for simplicity. Investments in each cost class has to stay below the cost class potential, $M_{p,i,cl}$, and $M_{q,i,i^*,cl}$, respectively, given by the consensus loop. Thus,

$$w_{i,p,cl} \leq M_{p,i,cl}, \quad \forall i \in I, \forall p \in P, \forall cl \in CL, \tag{A2}$$

$$h_{i,i^*,q,cl} \leq M_{q,i,i^*,cl} \quad \forall i, i^* \in I, \forall p \in P, \forall cl \in CL. \tag{A3}$$

The demand for electricity, $D_{i,t}$, has to be met in all regions at all times. Thus,

$$\sum_{p \in P^{gen}} g_{i,p,t_{s,r}} \geq D_{i,t_{s,r}} + \sum_{i^* \in I \setminus i} e_{i^*,i,t_{s,r}} + \sum_{p \in P^{bat}} \left( b_{i,p,t_{s,r}}^{charge} - b_{i,p,t_{s,r}}^{discharge} \right) + d_{i,t_{s,r}}^{hydrogen}, \forall i \in I, \forall t_s \in T_s, \ \forall s \in S \tag{A4}$$

where $e_{i^*i,t_{s,r}}$ is the exported electricity from region i to region i* (import is negative export), $b_{i,p,t_{s,r}}^{charge}$ is the charging of battery technology p and $b_{i,p,t_{s,r}}^{discharge}$ is the discharging of the same, and $d_{i,t_{s,r}}^{hydrogen}$ is the electricity demand for hydrogen production to industry.

Import and export is required to be balanced and export must be less than installed transmission capacity.

$$e_{i^*,i,t_{s,r}} = -e_{i,i^*t_{s,r}}, \quad \forall i, i^* \in I, \forall t_{s,r} \in T_s, \ \forall s \in S, \tag{A5}$$

$$e_{i^*,i,t_{s,r}} \leq \sum_{i^* \in I \setminus i} \sum_{q \in Q} \sum_{cl \in CL} h_{i,i^*,q,cl}, \quad \forall i, i^* \in I, \ \forall t_{s,r} \in T_s, \ \forall s \in S. \tag{A6}$$

The level of generation has to stay below the installed capacity, weighted by profile, $W_{i,p,t_{s,r}}$, which is weather-dependent for wind and solar power (but always equal to 1 for the thermal technologies).

$$g_{i,p,t_{s,r}} \leq \sum_{cl \in CL} w_{i,p,cl} \cdot W_{i,p,t_{s,r}}, \forall i \in I, \forall t_{s,r} \in T_s, \ \forall s \in S, \forall p \in P \tag{A7}$$

Flow batteries and Lithium Ion batteries are amongst the investment options in the model. For each storage there is an energy balance constraint controlling the storage state:

$$g_{i,p,t_{s,r+1}} \leq g_{i,p,ts,r} + \eta_p b_{i,p,t_{s,r}}^{charge} - b_{i,p,t_{s,r}}^{discharge}, \forall i \in I, \forall t_{s,r} \in T_s, \ \forall s \in S, \forall p \in P^{bat}. \tag{A8}$$

The charging and discharging of batteries are required to be less than the installed battery storage capacity $\sum_{cl \in CL} w_{i,p^{bat\_storage},cl}$, assuming a C-factor of 1.

For the cases including hydrogen demand and hydrogen storage, there is a hydrogen balance equation assuring that a constant demand for hydrogen from industry, $D_i^{hydrogen}$, is met by hydrogen production in the electrolyzer, $\eta^{electrolysis} d_{i,t_{s,r}}^{hydrogen}$:

$$g_{i,p,t_{s,r+1}} \leq g_{i,p,t_{s,r}} + \eta^{electrolysis} d_{i,t_{s,r}}^{hydrogen} - D_i^{hydrogen}, \forall i \in I, \forall t_{s,r} \in T_s, \ \forall s \in S, \forall p \in P^{hydrogen} \tag{A9}$$

where the electricity consumption in the electrolyzer $d_{i,t}^{hydrogen}$ is limited by the electrolyzer capacity $\sum_{cl \in CL} i_{i,p^{hydrogen},cl}$ and $\eta^{electrolysis}$ is the efficiency of the electrolysis process.

Investments in wind and solar power capacity cannot exceed the regional resources for the respective technology, $A_{i,p}$. For onshore wind, sites are ordered into classes depending on wind conditions, and there is a resource constraint for every class. Offshore wind pertains to its own class.

$$\sum_{cl \in CL} w_{i,p,cl} \leq A_{i,p}, \forall i \in I, \forall p \in P^{wind} \tag{A10}$$

where $P^{wind}$ is the set of wind classes. For solar power, there is a total resource constraint for the modeled region $i$.

$$\sum_{cl \in CL} \sum_{p \in P^{solar}} l_{i,p,cl} \leq \sum_{p \in P^{solar}} A_{i,p}, \ \forall i \in I \tag{A11}$$

Previous work has shown that the inclusion of thermal cycling has a substantial impact on the cost-optimal electricity system composition [44]. Thermal cycling is here accounted for by applying the relaxed unit commitment approach suggested by [24] and evaluated relative to the full-unit commitment by the author [25]. With this approach, there is a separate variable for capacity, which is active and available for generation in each technology aggregate for each-time step, $g_{p,t_{s,r}}^{active}$. Thus,

$$g_{i,p,t_{s,r}} \leq g_{i,p,t_{s,r}}^{active}, \forall i \in I, \forall t_{s,r} \in T_s, \ \forall s \in S, \forall p \in P^{thermal}. \tag{A12}$$

The active capacity is used to assure that the level of generation is higher than the minimum load level of the active part of the technology aggregate, $L_p^{min}$ :

$$L_p^{min} g_{i,p,t_{s,r}}^{active} \leq g_{i,p,t_{s,r}}, \forall i \in I, \forall t_{s,r} \in T_s, \ \forall s \in S, \ \forall p \in P^{thermal}. \tag{A13}$$

The amount of capacity started is controlled by the variable $g_{i,p,t_{s,r}}^{on}$:

$$g_{i,p,t_{s,r}}^{on} \geq g_{i,p,t_{s,r}}^{active} - g_{i,p,t_{s,r-1}}^{active}, \forall i \in I, \forall t_{s,r} \in T_s, \ \forall s \in S, \ \forall p \in P^{thermal}. \tag{A14}$$

The start-up cost is proportional to started capacity $g_{i,p,t_{s,r}}^{on}$ and the part-load cost is proportional to the difference between the active capacity and generation level:

$$c_{i,p,t_{s,r}}^{cycl} \geq g_{i,p,t_{s,r}}^{on} \cdot C_{i,p,t_{s,r}}^{on} + \left( g_{i,p,t_{s,r}}^{active} - g_{i,p,t_{s,r}} \right) C_{i,p,t_{s,r}}^{part}, \forall i \in I, \forall t_{s,r} \in T_s \backslash t_{s,|T|}, \forall s \in S, \forall p \in P^{thermal}. \tag{A15}$$

To avoid boundary effects, a value for the thermal generation in operation during the final hour of the 2-week segment is applied that is proportional to the start-up cost paid in the first hour of the 2 two-week segment, based on the capacity started $G_{i,p,t_{s,1}}^{on}$ and active $G_{i,p,t_{s,1}}^{active}$ in the first hour of the next season given by the previous iteration.

$$c_{i,p,t_{s,r}}^{cycl} \geq g_{i,p,t_{s,r}}^{on} \cdot C_{i,p,t_{s,|T|}}^{on} + \left( g_{i,p,t_{s,|T|}}^{active} - g_{i,p,t_{s,|T|}} \right) C_{i,p,t_{s,|T|}}^{part} - g_{i,p,t_{s,|T|}}^{active} \frac{0.5 C_{i,p,t_{s,1}}^{on} G_{i,p,t_{s,1}}^{on}}{G_{i,p,t_{s,1}}^{active}}, \ \forall i \in I, \ \forall s \in S, \ \forall p \in P^{thermal} \tag{A16}$$

Thus, if thermal capacity is active in the end of one 2-week segment and also in the beginning of the subsequent 2-week segment, the start-up cost for that capacity is share equally between seasons. Once capacity is deactivated, it cannot become active again during the interval K, which encompasses the time-steps k in the start-up interval:

$$g_{i,p,t_{s,r}}^{on} \leq \sum_{cl \in CL} w_{i,p,cl} - g_{i,p,t_{s,r-k}}^{active}, \forall i \in I, \ \forall s \in S, \forall k \in K. \tag{A17}$$

When the investment problem has been solved for the 26 2-week segments, information on investments in different types of generation, transmission, and variation management capacity in each 2-week segment is collected to form one capacity cost-curve for each technology and region in the consensus loop. In the initial solve, all 2-week segments share the investment cost equally, i.e., capacity is weighted by 1/26.

The capacity cost-curves are composed of 26 steps, where the length of the first step corresponds to the capacity investment level common to all 26 2-week segments. The length of the second step represents capacity investment additional to the first step shared by all the 2-week segments except one,

and so on. In order to determine the lengths of the steps, the number of 2-week segments $R_{p,i,s}$ that have a lower level of installed capacity of technology p in region i than the 2-week period s is calculated as:

$$R_{p,i,s} = 1 + |S| - \sum_{s^* \in S} \left[ i_{p,i,s} \leq i_{p,i,s^*} \right], \ \forall p \in P, \forall i \in I, \ \forall s \in S \tag{A18}$$

where S is the set of 2-week periods. The Iverson brackets are here applied, returning "1" if the expression within holds but otherwise returning "0". It follows that the length of the first step in the cost- curve $M_{p,i,cl_1}$ is given by:

$$M_{p,i,cl_1} = \frac{\sum_{s \in S} \left[ R_{p,i,s} = 1 \right] i_{p,i,s}}{\sum_{s \in S} \left[ R_{p,i,s} = 1 \right]} \forall p \in P, \forall i \in I \tag{A19}$$

where cl1 is the first element in the set of cost classes CL. The lengths of the subsequent steps in the cost-curve are calculated sequentially as follows:

$$M_{p,i,cl_m} = \frac{\sum_{s \in S} \left[ R_{p,i,s} = m \right] i_{p,i,s}}{\sum_{s \in S} \left[ R_{p,i,s} = m \right]} - \sum_{n=1}^{m-1} M_{p,i,cl_n}, \ \forall p \in P, \forall i \in I, \ \forall m \in \{2, \dots, |CL|\}. \tag{A20}$$

The length of the last step in the cost-curve is set to be very large, i.e., three-times the maximum annual load in the respective region. The height of each step in the cost-supply curve, i.e., the weight of the investment, is given by the number of 2-week segments sharing the investment:

$$C_{cl_m}^{share} = \frac{1}{|S| - (n - 1)}, \ \forall m \in \{1, \dots, |CL|\}. \tag{A21}$$

This cost is slightly modified in two ways: 1) the cost share is lower in the first iterations to enable the capacity with high investment costs to stabilize before extinction and 2) the cost share is lower for those 2-week periods that have not invested in capacity, which other 2-week periods have. This "rebate" is reduced with the iteration number. Thus,

$$C_{i,p,s,cl_m}^{share} = \frac{a_{j,p,i,s}}{\left( |S| - b_j(m - 1) \right)}, \ \forall m \in \{1, \dots, |CL|\}, \forall p \in P, \forall i \in I, \ \forall s \in S, \ j \in \{1, \dots, 10\}. \tag{A22}$$

Table A3 lists the choices for scalars a and b for each iteration j, where a can take on a high or low value depending on if investments have been made for that technology, region and two-week period (p,i,s).

**Table A3.** Coefficients in the consensus loop.

| Iteration Number (j) | $a_j^{low}$ | $b_j$ | $a_j^{high}$ |
|:---:|:---:|:---:|:---:|
| 1 | 0.5 | 0.5 | 0.1 |
| 2 | 0.6 | 0.6 | 0.1 |
| 3 | 0.7 | 0.7 | 0.2 |
| 4 | 0.8 | 0.8 | 0.2 |
| 5 | 0.8 | 0.9 | 0.3 |
| 6 | 0.8 | 1 | 0.4 |
| 7 | 0.8 | 1 | 0.5 |
| 8 | 0.8 | 1 | 0.6 |
| >8 | 0.8 | 1 | 0.6 |

For scenarios with gradually increasing costs for generation capacity or operation over the years, this increase is likely to impact investments and needs to be transferred to prior years. Electricity

generation technologies that rely on fossil fuels are for example typically subject to a gradual increase in operational costs over the decades considered, which reduces the cost-competitiveness of these technologies in the long-term perspective. Assuming that the total cost for investments and operation of a power plant is evenly distributed across all of its hours of operation, some of the operational costs from later years need to be shifted to earlier years. The net present value of these future operational cost (with interest r, num(y) as the numerical value of year y) is added to the objective function. Thus, we define the additional operational costs, $C_{p,t_s}^{add}$, as:

$$C_{p,t_{s,r},y^{inv}}^{add} = \frac{1}{\left|Y_p^{life}\right|} \sum_{y \in Y_p^{life}} \frac{1}{(1+r)^{(num(y)-num(y^{inv}))}} \left(C_{p,t_{s,r}}^{run,y} - C_{p,t_{s,r}}^{run,y^{inv}}\right), \forall t_{s,r} \in T_s, \ \forall s \in S, \ \forall p \in P, \forall y \in Y \quad (A23)$$

where $y^{inv}$ is the investment year and $Y_p^{life}$ is the set of years within the lifetime of technology p invested in the year $y^{inv}$. The denotation $num(y)$ here indicates the numerical value of the set element $y$. The added operational cost is added to the running cost, $C_{p,t_{s,r}}^{run}$, in the objective function (A1) for respective investment year.

**Appendix D**

The spatial segmentation of the building stocks presented in Mata et al. [31] is based on climate regions, and do not follow the spatial segmentation used in this paper. Thus, to represent the building stock on the segmentation seen in Figure A1 the original spatial segmentation needs to be adapted. This adaptation is done through connecting EU-buildings statistics on the number of buildings on a Nomenclature of Territorial Units for Statistics (NUTS)-2 level, taken from the Eurostat dataset "Conventional dwellings by occupancy status, type of building and NUTS 3 region"[45], to both the original spatial segmentation from Mata et al. and the regions in Figure A1, thereby translating the original segmentation to the segmentation used in this paper. The connection between the original spatial segmentation, NUTS-2 regions, and scenario regions is done in the following steps:

1.  Assign NUTS-2 regions to the climate regions deployed in [31]. As the climate regions are not based on a NUTS division the mapping between the regions is not perfect, i.e., borders of a climate region do not align with NUTS-2 region borders. In cases where the NUTS-2 region overlaps two climate regions the NUTS-2 region is assigned to the climate region in which it has the largest area.
2.  Segment the original building stock and the NUTS-2 data on number of buildings (coming from EU statistics) into archetype categories of single-family dwellings (SFDs) and multi-family dwellings (MFDs). In the original building stock representation this division has already been made. For the Eurostat NUTS-2 building statistics the categories RES1 and RES2 are assigned as SFD and RES_GE3 are assigned as MFDs.
3.  Create weights, i.e., the number of buildings, for each archetype building in each NUTS-2 region. Start by summing up the total number of SFDs and MFDs, separately, from the Eurostat data in all the NUTS-2 regions belonging to a specific climate region. Then calculate the share of SFDs and MFDs in each of these NUTS-2 in relation to the calculated total number of SFDs and MFDs. This gives the distribution of SFDs and MFDs for the NUTS2-resions within each climate region. The weight of each archetype building in the original building stock description, which is for a whole climate region, is then divided into the NUTS-2 regions based on the share of the category (SFD or MFD) that the archetype building belongs to in the corresponding NUTS-2 region. Thereby, creating weights for each archetype building in each NUTS-2 region.
4.  The final step is assigning the archetype weights from each NUTS-2 region to each region in Figure A1. As these regions correspond to specific NUTS-2 regions the mapping between these is straight forward. The weights for each archetype for all NUTS-2 regions belonging to a region is summed up, resulting in one weight for each archetype in each region in Figure A1.

As mentioned in Section 2.4 solar irradiation and temperature profiles are needed for modelling of the space heating demand. Thus, the building stock in each scenario region is assigned a solar irradiation and a temperature profile with a temporal resolution of 1 hour. The data for these irradiation and temperature profiles are taken from MERRA-2 [46,47]. The profiles for each region in Figure A1 are created through weighting together the profiles for each NUTS2 region belonging to a region. The NUTS2 profiles are weighted according to the number of buildings in each NUTS2 region. The profiles for each NUTS2 region are in turn created by assigning them the profiles from the weather data set that are closest to the center point of the region.

As mentioned in Section 2.4, the refurbishments of the buildings follow the reports on cost optimal levels from the EU. These are implemented by assigning new U-values for each archetype building in accordance with the reports. Furthermore, new buildings are assumed to have a negligible impact on the future demand both due to the minor amount of new buildings in relation to the standing buildings stock and due to that all new buildings in the EU are required to be nearly zero-energy buildings by 2020 [48]. Hot water demand is assumed to be constant over the year. It is assumed that the change in energy demand, and thus the refurbishments, follow a sigmoid curve over the modelling period, with 9%, 50%, 91%, and 100% of the total energy demand from the refurbished/new buildings stock reached at years, 2030, 2040, 2050, and 2060, respectively.

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
