# Peer review of "The Benefit of Collaboration in the North European Electricity System Transition—System and Sector Perspectives"

_energies, doi:10.3390/en12244648_

Round 1

Reviewer 1 Report

Summary of Paper

This paper aims to investigate the electrification of the heat, power and transport sectors with flexibility provision on both electricity system investments and electricity prices. The authors also would like to examine the role of collaboration among the sectors in reducing the system costs of this electrification.

An H2D model is used to determine the investment decisions over multiple 2-week segments. It is not clear to me why 2-week segments are an advantage; this would also appear to be a very short time-frame for investment decisions to be made. I do not understand the relationship between the 2-week segments and this is not clear to me from the model explanation.

Potential Contribution

Although the topic of the interrelationship between electrification of sectors and flexibility provision is highly important, there is confusion about the scope of the paper - a large number of different Technologies, geographical regions and sectors are mentioned and then only briefly covered in the main body of the paper.

Strengths

The collaboration among sectors in the integration of renewable power is a crucial topic and the paper makes interesting findings in relation to the potential contribution of electric vehciles in reducing average power prices and absorbing excess power. This role of electric vehicles is possibly where the main story of the paper lies. What is very interesting is the potential interrelationship between countries - i.e. the fact that vehicles charging in the UK could help the absorption of German solar power. It is clear that the authors have created a detailed model of the different sectors including a wide variety of technologies; it is just that this model is not presented well.

There is much less coverage of the contribution of the steel sector and heat pumps though.

Weaknesses

The authors state that collaboration leads to a fall in system costs of 8%, but a systematic explanation of how this occurs is absent - a big contribution comes from electric vehicle charging, but how important is it compared to the other sectors in leading to this reduction in system costs?

Technologies such as coal with CCS are discussed in relation to Germany, yet coal power will be phased out by 2038, so there are limited prospects, under the current political framework, for CCS deployment in the coal sector. The authors do acknowledge this at the end of the paper, but they need to justify why coal with CCS is included in the model in the first place.

The model includes 12 European countries, but there is limited focus on the interrelationship between these countries and findings often focus on just one or two countries, with little coverage of the others.

Assumptions are sometimes not justified in the text - why, for example, one year is taken to represent a decade. This needs to be supported. Secondly, what is the justification for a CO2 price of 400 €/tonne of CO2 in 2050? How is this figure derived?

Furthermore, where do the figures for the annual electricity demand in future decades for the transport sector and heat sector come from? Are they outcomes from the modelling or do they come from other sources?

The conclusions are insufficiently developed and are in the form of bullet points.

Reviewer 2 Report

The Authors present a consistent model where many scenarios can be considered. They present its limitations and they realize explicitly limitation in forecasting. That's a disadvantage of all complicated model with long analyzed period and uncertain inputs. The big advantage of the paper is focusing on demand increase wich has to be matched by production (supply).

Some specific remarks

Lines 26-34. Increases given in % do not refer to periods. (yearly?; during the decade?)

Figure 3. Lines are hard to read

Figure 4. To much information on one figure make it uninformative, to small axis descriptions

Round 2

Reviewer 1 Report

The Points made in the Initial Review have now been addressed